# Characteristic Immune Dynamics in COVID-19 Patients with Cardiac Dysfunction

**DOI:** 10.3390/jcm11071880

**Published:** 2022-03-28

**Authors:** Filipe André Gonzalez, Miguel Ângelo-Dias, Catarina Martins, Rui Gomes, Jacobo Bacariza, Antero Fernandes, Luís Miguel Borrego, EchoCrit Group

**Affiliations:** 1Intensive Care Department, Hospital Garcia de Orta, 2805-267 Almada, Portugal; rui.gomes@hgo.min-saude.pt (R.G.); jacobo.blanco@hgo.min-saude.pt (J.B.); antero.fernandes@hgo.min-saude.pt (A.F.); 2CEDOC—Chronic Diseases Research Center, NOVA Medical School, NOVA University of Lisbon, 1099-085 Lisbon, Portugal; miguel.dias@nms.unl.pt (M.Â.-D.); catarina.martins@nms.unl.pt (C.M.); borregolm@gmail.com (L.M.B.); 3Immunoallergy Department, Hospital da Luz Lisboa, 1500-650 Lisbon, Portugal

**Keywords:** COVID-19, speckle-tracking echocardiography, diastolic dysfunction, flow cytometry

## Abstract

Background: We aimed to explore immune parameters in COVID-19 patients admitted to the intensive care unit (ICU) to identify distinctive features in patients with cardiac injury. Methods: A total of 30 COVID-19 patients >18 years admitted to the ICU were studied on days D1, D3 and D7 after admission. Cardiac function was assessed using speckle-tracking echocardiography (STE). Peripheral blood immunophenotyping, cardiac (pro-BNP; troponin) and inflammatory biomarkers were simultaneously evaluated. Results: Cardiac dysfunction (DYS) was detected by STE in 73% of patients: 40% left ventricle (LV) systolic dysfunction, 60% LV diastolic dysfunction, 37% right ventricle systolic dysfunction. High-sensitivity cardiac troponin (hs-cTn) was detectable in 43.3% of the patients with a median value of 13.00 ng/L. There were no significant differences between DYS and nDYS patients regarding mortality, organ dysfunction, cardiac (including hs-cTn) or inflammatory biomarkers. Patients with DYS showed persistently lower lymphocyte counts (median 896 [661–1837] cells/µL vs. 2141 [924–3306] cells/µL, *p* = 0.058), activated CD3 (median 85 [66–170] cells/µL vs. 186 [142–259] cells/µL, *p* = 0.047) and CD4 T cells (median 33 [28–40] cells/µL vs. 63 [48–79] cells/µL, *p* = 0.005), and higher effector memory T cells (TEM) at baseline (CD4%: 10.9 [6.4–19.2] vs. 5.9 [4.2–12.8], *p* = 0.025; CD8%: 15.7 [7.9–22.8] vs. 8.1 [7.7–13.7], *p* = 0.035; CD8 counts: 40 cells/µL [17–61] vs. 10 cells/µL [7–17], *p* = 0.011) than patients without cardiac dysfunction. Conclusion: Our study suggests an association between the immunological trait and cardiac dysfunction in severe COVID-19 patients.

## 1. Introduction

Severe acute respiratory syndrome coronavirus 2 (SARS-CoV-2) causes coronavirus disease 2019 (COVID-19). Although the predominant manifestation of COVID-19 is acute respiratory distress syndrome (ARDS), infection with SARS-CoV-2 is also associated with multiple cardiac manifestations [1]. In severe COVID-19 patients, diagnostic transthoracic echocardiography (TTE) allows early recognition of cardiac injury and impacts clinical management, reducing organ dysfunction and mortality [2,3]. Importantly, the expression of cardiovascular disease seems to be a marker of a poor prognosis in COVID-19 [1]. Viral infection, in general, has been previously identified as a cause of myocarditis that is usually defined by evidence of inflammation in the heart. It has also been recognized that infectious viral heart disease forms may not be associated with the typical inflammatory infiltrate [4]. Given the large number of viruses that can cause myocarditis [4], among which we found other coronaviruses, it is rational to hypothesize that a novel coronavirus that causes cardiac injury may be doing so by causing myocarditis at least in some cases. The cardiac injury could occur due to direct viral-mediated cytopathic effects in the cardiac myocyte or by viral activation of immune mechanisms resulting in inflammatory cell infiltration [5].

Nevertheless, there are conflicting reports on the myocardial histology of patients with COVID-19 who had evidence of myocardial injury. A recent preliminary report on postmortem heart analysis demonstrated “gross cardiac enlargement” in 24 out of 25 cases, with evidence of left ventricular hypertrophy and moderate to marked atherosclerotic narrowing of the coronary arteries. Moreover, 60% of patients had evidence of a patchy epicardial mononuclear infiltrate with a predominance of CD4 T cells compared to CD8 T cells. Small vessel thrombi were observed in three cases, and one had hemophagocytosis within an area of epicardial inflammation [6].

Few studies have addressed the evolution of the immunologic profile in COVID-19 patients. Laing et al. [7] accomplished an exhaustive immunologic analysis, where they identified a core peripheral blood immune signature across 63 hospitalized patients with COVID-19 who were otherwise highly heterogeneous. The signature included discrete changes in B and myelomonocytic cell composition, profoundly altered T cell phenotypes, selective cytokine/chemokine upregulation and SARS-CoV-2-specific antibodies. Some signature traits identified links with other settings of immunoprotection and immunopathology. Others, including basophil and plasmacytoid dendritic cell depletion, were strongly correlated with the disease severity, while a third set of traits, including a triad of IP-10, interleukin-10 and interleukin-6, anticipated subsequent clinical progression. Moreover, another group studied the association between the immunologic profile and cardiac injury in COVID-19 patients, suggesting that the numbers of T and B lymphocytes were significantly decreased in the group with cardiac injury. Still, the reduced proportion of B lymphocytes did not differ [8].

Our research focused on the immunologic profile evolution in COVID-19 patients with cardiac dysfunction (and not only injury detected by biomarkers), considering the scarcity of data on the underlying immune mechanisms related to cardiovascular disease in this context. Furthermore, we sought to determine if a specific immune activation in these patients could translate an underlying myocarditis-like mechanism driving the cardiac dysfunction. Hence, contingent upon independent validation in other COVID-19 cohorts, individual traits within this signature may collectively and individually guide treatment options, offer insights into COVID-19 pathogenesis and aid early, risk-based patient stratification that is particularly beneficial in phasic diseases such as COVID-19.

## 2. Materials and Methods

### 2.1. Study Design and Participants

This prospective observational study involves a comprehensive clinical and cellular characterization of COVID-19 patients over 18 years old, positive for SARS-CoV2 RT-PCR test, and admitted to the ICU. All patients were enrolled consecutively between June 2020 and September 2020 at Hospital Garcia de Orta-Almada (Portugal). Exclusion criteria comprised type 1 acute myocardial infarction and pulmonary embolism as presenting diagnosis and inappropriate acoustic window for STE.

Previous studies have shown the prognostic value of performing serial assessments of biomarkers at early disease onset [9]. Accordingly, in our study, patients were assessed at the earliest opportunity following admission (<24 h–D1), from the second to the third day (D3), and from the seventh to the tenth day (D7).

Comprehensive STE was performed by an intensive care and echocardiography specialist at each time point. Besides evaluating invasive arterial pressure and central venous pressure, peripheral blood samples were collected to assess cardiac and inflammatory biomarkers, along with an immunophenotyping characterization by multiparametric flow cytometry. The criteria to choose these particular timepoints is related to the physiopathology and natural history of other cardiac dysfunctions induced by infection, such as septic-induced myocardial dysfunction (previously known as septic cardiomyopathy) [10,11].

Additionally, data on other markers were collected from the hospital patient file, including C-reactive protein (CRP), procalcitonin (PCT), ferritin, erythrocyte sedimentation rate (ESR), D-dimers, fibrinogen and complete blood count with platelets.

All clinical data and medication were analyzed, namely demographic data: gender, age, previous history of hypertension, diabetes mellitus, chronic obstructive pulmonary disease (COPD), body mass index (BMI), and current medication with remdesivir or corticosteroid. Clinical outcomes such as mechanical ventilation, P/F ratio (oxygen partial pressure to inspired oxygen fraction ratio), shock, acute kidney injury, ICU and hospital length of stay, and mortality were also recorded. Besides cardiovascular risk factors, baseline cardiac function by echocardiography was searched for in the previous history through family physician files.

### 2.2. Echocardiography

All echocardiographic measures were obtained with simultaneous EKG display, and the three different sets of measures were obtained with the patient in the same position. All Doppler measurements were averaged from three measurements in sinus rhythm and ten measurements in atrial fibrillation. In addition to the qualitative examination of chambers and valves, the following measures were obtained via the standard parasternal and apical views: pulmonary artery systolic pressure; left atrium (LA) area and volume; left ventricle (LV) end-diastolic and end-systolic volumes (EDV and ESV) using biplane modified Simpson’s rule, from which the ejection fraction (EF) was calculated; right ventricle (RV) end-diastolic area and fractional area change (FAC); tricuspid annular plane systolic excursion (TAPSE); mitral annular plane systolic excursion (MAPSE) and left ventricle outflow tract (LVOT) diameter, LVOT VTI measured on pulsed-wave Doppler from which stroke volume (SV) was calculated, and then cardiac output (CO) and cardiac index (CI) derived. Mitral flow measurements included: E maximal velocity, E deceleration time and A maximal velocity. Tissue Doppler measurements were: RV S’ wave, LV septal e’, lateral e’ and systolic myocardial velocity during ejection (Sa).

To determine LV, RV and LA strain, four-chamber views were acquired after optimizing the sector size, gain, depth, compress and time-gain compensation. The frame rate was maximized (76 ± 24 fps) by decreasing the depth and sector width. The region of interest was manually traced at ED along the endocardial border. The software then automatically tracked the endocardial contours throughout the cardiac cycle. Manual adjustments were made to the contours as needed to optimize tracking. Images were analyzed offline using dedicated software (Toshiba Xario 200, 2D Wall Motion Tracking (WMT) software). The peak strain value was derived from the maximal inflection point on the composite LA strain curve and graded based on recently published cutoffs [12].

The cardiac dysfunction was classified according to the European Society of Cardiology (ESC) and American Society of Echocardiography (ASE) guidelines for chamber quantification classification [13]. LV systolic dysfunction was defined as a calculated LVEF < 50% or LVGLS < 20%. RV systolic dysfunction was defined as RV FAC < 35%, TAPSE < 17 mm, S’ < 9.5 cm/s or RVGLS < 20%. Classic criteria classified LV diastolic dysfunction according to 2016 ASE/EACVI guidelines (Section A.1, i and ii—Diastolic dysfunction classification) [14] and by LA strain to assess contractile (LA-CT < 10%) and reservoir (LA-RV < 30%) functionality using definitions from previous studies [12]. Repeated measures for different echocardiographic parameters were taken by the same physician and by two different physicians to assess inter and intraobserver variability.

Venous ultrasound was performed to assess inferior vena cava (IVC) variability ((Max-Min)/Max) measured in M-mode, and portal vein pulsatility index (PVPi) was measured by pulsed-wave Doppler in the portal vein ((Max-Min)/Max), according to the previous classification of venous congestion by ultrasound [15].

### 2.3. Cardiac and Inflammatory Biomarkers

Peripheral blood samples were collected into tubes without an anticoagulant agent and were centrifuged after coagulum retraction. Serum samples were then separated, immediately tested for cardiac biomarkers, or further aliquoted and stored at −20 °C for inflammatory biomarkers analysis.

As for cardiac markers, high sensitive troponin T (reference range < 13 ng/L) and NT-proBNP (reference range < 125 pg/mL) were assessed by immunoassay with electrochemiluminescence technology (or “ECLIA”) in Cobas (Roche) analyzers.

To quantify serum levels of IL-1β, IL-6 and adrenomedullin (ADM), commercial immunoassays were used according to the procedures defined by the respective manufacturers. The Quantikine^®^ELISA kits (R&D Minneapolis, MN, USA) were used for IL-1β and IL-6, presenting sensitivities of 1 pg/mL (assay range 3.9–250 pg/mL) and 0.7 pg/mL (assay range: 3.1–300 pg/mL), respectively. Low, medium and high control samples were run parallel at each series to assure curve verification. For ADM measurement, the RayBio^®^ Human/Mouse/Rat Adrenomedullin Enzyme Immunoassay Kit was used (RayBiotech, Norcross, GA, USA). The assay has a sensitivity of 0.9 ng/mL (assay range 0.1–1.000 ng/mL). The SigmaPlot software version 14.5 (Systat Software Inc. (SSI), San Jose, CA, USA) was used to perform a four-parameter logistic regression curve and determine final ADM concentrations. Samples were run in duplicates (accepted whenever the coefficient of variation was below 15%), and the mean values were used for each sample.

### 2.4. Flow Cytometry

For immunophenotyping characterization, peripheral blood samples collected into EDTA tubes were analyzed within 24 h after collection. In brief, cells were incubated with a prevalidated panel of monoclonal antibodies for 15 min, lysed, washed, and acquired in an 8-color BD FACS Canto II Flow Cytometer. The panel of monoclonal antibodies included CD3, CD4, CD8, CD16, CD19, CD20, CD27, CD38, CD45, CD45RA, HLA-DR, IgD, IgM and TCR gamma delta, assuring the identification of several leukocyte populations, and particularly, distinct subsets of T cells (i.e., naïve, memory and activated T cells within CD4, CD8 and gamma delta T cells) and B cells (i.e., naïve, unswitched and switched memory, and plasmablasts). A single-platform strategy with BD Trucount tubes was also used to assure absolute cell counts for each subset addressed.

BD FACS Diva software was used for acquisition purposes, and Infinicyt 2.0 software (Cytognos, Salamanca, Spain) was used for data analysis. All gating strategies are presented in (Section A.2—Gating strategy for identifying immune subsets).

### 2.5. Statistics

All statistical analysis was performed with GraphPad Prism 9 software v9.0.2 (Graph Pad, San Diego, CA, USA). Categorical variables were presented as percentages and analyzed with Fisher’s exact test or Qui Square test as applicable. Continuous variables were expressed as mean and standard deviations (SDs) or median and interquartile range (IQR), depending on the normality of distributions. Group comparisons were performed using the unpaired t-test with Welch’s correction or the nonparametric Mann–Whitney U-test. Three or more paired groups were compared with Friedman’s test, followed by the Dunn multiple comparison test. The nonparametric Spearman correlation coefficient assessed correlation studies. Statistical significance was considered for *p* < 0.05.

### 2.6. Ethics

This was a prospective observational study on current practice, and all collected data were the standard monitoring and intervention data used in critical care. All monitoring and intervention attitudes have been extensively described in the literature, and patients were assessed and treated routinely. Data were anonymously collected, and no information allowing patient identification was collected. Patient anonymity was maintained both during data processing and publishing. A patient number was used in all echocardiograms recorded for later review by an expert. According to local legislation, consent from the patient or next of kin to record data and draw blood was obtained before study enrollment. The Ethics Committees approved the study of Hospital Garcia de Orta and NOVA Medical School (CE no. 44-2020-CEFCM).

## 3. Results

### 3.1. Clinical Characterization

Between June and September of 2020, 40 patients met the inclusion criteria, from which 10 were excluded (2 had a myocardial infarction, 2 had a pulmonary embolism, and 6 had an inappropriate acoustic window). The baseline characteristics of all patients also separated in patients with (DYS) and without (nDYS) cardiac dysfunction at ICU admission are shown in Table 1.

Of the final 30 patients included, 63% were male with a mean age of 60.7 ± 14.8 years. Hypertension was the most common risk factor (73.3%), followed by obesity (40%) with a mean BMI of 31 kg/m^2^, and half of the patients were under ACEi or ARA treatment. The family physicians’ previous history files showed that none of the patients had previous cardiac dysfunction detected by echocardiography, except three that had no registries. Overall, 22 patients (73.3%) had some cardiac dysfunction detected by STE, either left ventricle (LV) systolic dysfunction (40%), LV diastolic dysfunction (63.3%) or right ventricle (RV) systolic dysfunction (36.7%). Hypotension, acute kidney injury or mechanical ventilation occurred in 23.3% of patients. Five patients (16.7%) died during the study. Troponin was detectable in 43.3% of the patients with a median value of 13.00 ng/L, while NT-proBNP had a median value of 275 pg/mL. In addition, there were no significant differences between DYS and nDYS patients regarding mortality, organ dysfunction and cardiac or inflammatory biomarkers (including troponin T, NT-ProBNP, IL-1β, IL-6, ADM, but also CRP, PCT, ferritin, and ESR).

### 3.2. Echocardiographic Characterization

From the STE evaluation, 73.3% of patients demonstrated a cardiac dysfunction at ICU admission, and 80% of patients experienced a cardiac dysfunction within the first seven days of ICU stay (Table 2) (Appendix A).

Speckle-tracking measures were similar to the 2D measures for both LV and RV systolic function, mainly LV GLS vs. LVEF, MAPSE, LV S’ and RV GLS vs. RV FAC, TAPSE, RV S’. Significantly different results for LV diastolic function were seen. Namely, assessment of LA strain markedly improved LV diastolic dysfunction detection compared to the classic criteria (60% vs. 23.3%, respectively; *p* = 0.008). In addition, the DYS group presented lower median values of both LA RV (23.1% vs. 39.1%; *p* = 0.008) and LA CT (7.7% vs. 13.9; *p* = 0.0002) compared to the nDYS group, while no differences were observed regarding classic measurements, including E/A, e’, E/e’ and sPAP. Venous congestion evaluation (IVC% and PVP) did not differ significantly between groups with organ dysfunction such as AKI, P/F or mechanical ventilation. Nevertheless, PVP showed a stronger correlation with the congestion biomarker NT-proBNP than IVC% (PVP: r = 0.32, *p* = 0.005; IVC%: r = 0.05, *p* = 0.640), considering patients in both groups and all time points. Inter- and intraobserver variability was calculated for LVEF, LV GLS, RV FAC, RV GLS, TAPSE, E/A, e’, E/e’, sPAP and LA volume. LA CT and LA RV, ranged from 0.8–1.8% ± 0.9–1.1% for interobserver and 0.4–1.1% ± 0.3–0.9% for intraobserver variability. Specifically for STE measurements: for the interobserver variability (median (IQR 25–75)): LV GLS 0.8 (0.5–1.1), RV GLS 0.7 (0.5–1.1), LA RV 1.3 (0.9–1.9), LA CT 1.1 (0.8–1.7); for the intra-observer variability: LV GLS 0.6 (0.5–0.8), RV GLS 0.7 (0.5–0.8), LA RV 1.3 (0.5–0.9), LA CT 0.9 (0.4–0.9).

### 3.3. Immunological Characterization

In general, patients presented with a normal leukocyte count at admission, and despite a lower limit count for total lymphocytes (median 957 cells/µL), there was a mild T lymphocytopenia (median 587 cells/µL and 56.86% of total lymphocytes). Nevertheless, T cell subsets were within the normal reference ranges.

### 3.4. Longitudinal Characterization along with ICU Stay

To understand the variations in the clinical and immune profiles of patients with COVID-19 during their ICU stay according to their cardiac manifestations, we divided patients into two groups, depending on whether they showed any cardiac dysfunction at admission to the ICU. These groups were compared throughout the three time points. However, as some patients were discharged from the ICU or eventually died during this period, the number of patients assessed was not constant at all time points.

### 3.5. Dynamics of Cellular Immune Parameters

Considering the longitudinal assessment of immune data, both groups showed similar leucocyte and lymphocyte counts at admission (Table A1—Longitudinal dynamics of inflammatory, cardiac, and cellular parameters in DYS and nDYS patients). However, along all time points, DYS patients showed lymphocyte counts persistently in the lower limit of the normal range, while lymphocyte counts in nDYS patients recovered at D7. Activated (HLA-DR^+^) total T cell and CD4 T cell counts were similar between both groups at D1 and D3. However, nDYS patients showed an increase in the levels of both activated subsets at D7, compared to DYS patients (*p* = 0.046 and *p* = 0.005, respectively). In addition, the percentages of activated HLA-DR^+^ CD8 T cells also decreased at D3 in DYS patients (*p* = 0.037), increasing afterwards at D7 (*p* = 0.013), while nDYS showed similar values in all time points.

Regarding CD4 and CD8 T cell differentiation patterns, at admission, T_EM_ CD4 and CD8 T cells showed significantly increased percentages in the DYS group compared to nDYS (CD4 T_EM_: 10.87% vs. 5.92%; *p* = 0.025; CD8 T_EM_: 15.74% vs. 7.09%; *p* = 0.035) as shown in Figure 1.

Similar differences between groups were observed in D3 (*p* = 0.007) and D7 evaluations (*p* = 0.046) regarding CD4 T_EM_ cell percentages. Interestingly, the rates of CD8 T_EM_ cells decreased over time in both groups, though without statistical significance (*p* = 0.058).

### 3.6. Dynamics of Cardiac and Inflammatory Parameters

Considering other inflammatory biomarkers, DYS patients had lower platelets at D3, with slower recovery in platelet counts and CRP levels, and CRP decreased more at D7 in nDYS patients (*p* = 0.009). Although the IL-6 levels were not significantly different between groups, neither at baseline nor along the time, left atrial strain function parameters, such as LA RV and LA CT, correlated negatively with IL-6 levels at admission (LA CT, *p* = 0.0002, r = −0.63; LA RV, *p* = 0.009, r = −0.47), as displayed in Figure 2.

Interestingly, along the timeline, this correlation was either lost (for LA RV) or inverted (for LA CT) toward a positive correlation with IL-6 levels. Nevertheless, of the five nonsurvivors, three had the highest values of IL-6 (>300 pg/mL in two patients and another with 191 pg/mL), all at D7. Only one survivor had a similar peak of IL-6 (>300 pg/mL) at D3, decreasing afterwards at D7.

Cardiac function parameters did not parallel the evolution in the other immuno-inflammatory markers. During the three time points, ventricular and atrial, systolic and diastolic function measurements did not change significantly in both groups (Figure 3), and patients recovered clinically as translated by an increasing P/F ratio, mainly nDYS patients.

## 4. Discussion

To our knowledge, this is the first study to describe an association between the dynamic immunological and inflammatory profiles over time and cardiac dysfunction by STE, and not solely a cardiac injury defined by increased troponin levels in severe COVID-19 patients.

In the overall population, the clinical, imaging and routine laboratory findings of enrolled patients were similar to previously published data [7,8,16]. Patients with cardiac dysfunction showed persistently decreased lymphocyte counts, with lower recovery in the numbers of CD4 T, CD8 T and B cells as well as increased CD4 and CD8 central memory T (T_CM_) cells. Nevertheless, inflammatory markers were similar at admission in patients with and without cardiac dysfunction, and both groups had lymphopenia. In fact, lymphopenia has been described in COVID-19 patients with a different clinical spectrum of the disease [17], although the changes in different lymphoid subsets still need further clarification. Indeed, lymphocytopenia occurs in more than 80% of severe COVID-19 patients [18], and our results suggest that the degree of lymphocytopenia may reflect the severity of COVID-19 disease since the lymphocyte counts were persistently decreased in patients with cardiac dysfunction. As proposed for other markers such as troponin, early serial evaluations, not only the values at admission, can be helpful in patient monitoring, providing information on high-risk patients that could benefit from intensive and specific care [9]. In line with this, our data showed that CD4 and CD8 T cell subsets and B cells had poor recovery in the group with cardiac dysfunction, and we know these subsets, particularly T cells, are essential for the control of viral infection. Although the evolution of immune parameters within the first days of patient admission may have predictive value, the reason some patients develop and maintain lymphopenia remain unknown. Given that lymphocytes express low levels of angiotensin-converting enzyme 2 (ACE2), the cell entry receptor for both SARS-CoV-2 and SARS-CoV [19,20] the viral genome is rarely detectable in the peripheral blood of SARS-CoV-2 infected patients [21,22]. Therefore, it is reasonable to speculate that the decrease of peripheral lymphocytes can also result from mechanisms such as activation-induced cell death (AICD) or aggressive migration from peripheral blood to the lungs, where robust viral replication occurs [23]. Nevertheless, further studies are needed to elucidate the mechanisms underlying the observed lymphopenia and eventual migration patterns in SARS-CoV-2 infection or even their potential use as predictive markers.

Patients from our cohort with cardiac dysfunction showed a distinct T cell differentiation pattern, with a higher frequency of CD4 and CD8 T_CM_ and T_EM_ cells at admission and during the ICU stay. The striking loss of T cells, particularly of naïve CD4 T cells, has been already characterized in COVID-19 patients [24,25], though many effector and memory subsets are proportionally increased. Indeed, many T cell subsets proliferate during COVID-19, including those usually quiescent, such as T_CM_ and T_EM_ subsets [24], with 10-fold increases in blood CD4 and CD8 T_EM_ cells in G1 or S-G2/M cell cycle phases [7]. Furthermore, subsets may be increasing even though they are decreased in frequency [24], with proliferation rates likely to be influenced by disease stage and severity and even enhanced in patients with severe disease [26]. Additionally, the increased presence of effector cells in the DYS group may also reflect a later and less effective antigen clearance, with increased stimulatory signals and proliferation rates leading to exhaustion [27]. In contrast, patients without dysfunction may experience a reduction in antigen levels, consequently leading to increased apoptosis of antigen-specific effector cells in an attempt to achieve immune homeostasis. Thus, the impairment in the clearance of inflammatory signals and differentiated cells could be related to the status of cardiac dysfunction. Moreover, reduced antigen stimulation can also be an effect of the ongoing treatment with anti-inflammatory agents [28]. In brief, the underlying mechanisms for increased effector/memory T cells in DYS patients remain unclear, possibly resulting from variations in differentiation patterns, proliferation-apoptosis balance, and even migratory events between the periphery and the affected organs and tissues. Most likely, all these mechanisms might be conjugated [27].

Interestingly, although our patients had increased serum levels of IL-6, according to previous studies [7,8,16,29], these cytokine levels were not different between DYS and nDYS patients. Accordingly, circulating neutrophils and monocytes, major sources of IL-6 during inflammatory responses, showed no important variations between both groups. In fact, nDYS patients had an increase of circulating monocyte counts seven days after admission, with a significant decrease of serum PCR at the same time point, possibly supporting an earlier recovery of inflammation in nDYS (though IL-6 levels did not change accordingly). Moreover, IL-6 levels were not correlated with neutrophils and monocyte counts at any time point. Of course, we cannot obliviate other sources for this cytokine, such as endothelial cells, particularly when endothelial activation and disruption have been considered major drivers of severe COVID-19 [30]. Nonetheless, the absence of differences on IL-6 between DYS and nDYS can result from the impact of ongoing therapy since approximately half of the patients were under ACEi in both groups, and these agents are known to have an impact on both immune cells and serum cytokines such as IL-6 [31]. Still, left atrial strain function parameters, such as LA RV and LA CT—a more sensitive surrogate for diastolic dysfunction—correlated negatively with IL-6 levels at admission, which could help identify patients at risk for a protracted inflammatory state.

A previous systematic review defied the role of IL-6 in COVID-19 [32]. Although cytokine concentrations are elevated in patients with severe and critical COVID-19, the degree of hypercytokinaemia is markedly lower than in other disorders associated with elevated cytokines. Given these findings, the cytokine storm is problematic, and alternative mechanisms of COVID-19-induced organ dysfunction are worth considering. Several major randomized trials evaluating the use of interleukin (IL)-6 inhibitors or Janus kinase (JAK) inhibitors with or without corticosteroids in patients with COVID-19 suggest that adding a second immunomodulatory drug such as baricitinib or tocilizumab to dexamethasone provides clinical benefit in patients requiring oxygen supplementation [33,34]. The REMAP-CAP and RECOVERY trials, the two largest randomized controlled trials of tocilizumab to date, have reported a mortality benefit for tocilizumab among patients with rapid respiratory decompensation requiring oxygen delivery through a high-flow device or NIV [35,36]. In the COV-BARRIER trial, the difference in mortality was most pronounced in the subgroup of 370 patients receiving high-flow oxygen or NIV at baseline (17.5% in the baricitinib arm vs. 29.4% in the placebo arm; HR 0.52; 95% CI, 0.33–0.80; nominal *p* = 0.007). More broadly, the immune features of COVID-19 remain largely unsettled. However, caution is needed when drawing inferences about the underlying processes that such markers might reflect as well as their potential causal roles in disease.

This is the first study providing the incidence of cardiac dysfunction with multimodal description (including STE for the evaluation of LA strain as a surrogate of LV diastolic dysfunction) in association with an extensive sequential immunological analysis (in different time points) in critically ill COVID-19 patients. Most previous studies have defined cardiac injury only by troponin elevation [37,38], but troponin elevation does not necessarily reflect cardiac injury and is likely to be multifactorial [39]. In our study, troponin levels did not correlate with echocardiographic cardiac dysfunction as suggested in other studies [40]. Nevertheless, troponin is the most sensitive marker of myocardial injury and can translate either indirect myocardial injury, myocardial inflammation or myocarditis [41,42]. Given our cohort’s low troponin and NT-proBNP levels, we could hypothesize that an unbalanced proinflammatory milieu, rather than a direct cytopathic effect on myocytes, would lead to a cardio-depressor effect, which is just a speculating theory. Additionally, the absence of significant differences between the two groups in cardiac and inflammatory biomarkers (DD, CRP and ESV) could be further explained by the low severity disease, as shown by gravity scores, such as APACHE (median of ten in both groups) and SOFA (median of three in both groups), and similar organ dysfunction and mortality rates.

Nonetheless, in our cohort of critically ill COVID-19 patients, 73,3% of patients demonstrated a cardiac dysfunction at ICU admission, and 80% of patients experienced cardiac dysfunction within the first seven days of ICU stay. The most frequent abnormalities were LV diastolic dysfunction, followed by LV and RV systolic dysfunction. Our data regarding cardiac dysfunction prevalence in COVID-19 is in line with recent evidence suggesting that 70% of patients with COVID-19 harboured a cardiac injury within the first ICU admission, identified by multimodal cardiac assessment [43]. This incidence is higher than previous reports, with values ranging from 12% to 30% [37,38,44,45,46,47]. The higher incidence reported by Doyen and collaborators [43] could be explained by either the longitudinal cardiac assessment or the use of a much more sensitive tool to detect LV diastolic dysfunction, such as the STE [48], being simultaneously in line with incidences reported in critically ill patients with sepsis not related to COVID-19 [49]. Thus, it confirms that COVID-19 patients experienced more LV diastolic than systolic dysfunction [50].

Although LA strain is advocated to be less load-dependent [51], it is still influenced by age, gender, LV systolic and diastolic function [52], and probably by other less studied loading conditions in critical care, such as fluid or vasopressor overload, and the complex heart-lung interactions in mechanically ventilated patients [53]. Left and right ventricular systolic dysfunction measured by GLS had an incidence similar to other reports [54]. Although LV GLS was not significantly different from the other usual systolic function parameters, RV GLS was poorly correlated with FAC, TAPSE and RV S’, as previously demonstrated by our group [55]. Our cohort’s 36.7% incidence of RV dysfunction is in line with previous publications [54] and could be attributed to the parenchymal or microvascular disease in COVID-19. However, this never presented as hypotension or cardiogenic shock, which could be explained by the only moderate grade of RV dysfunction (median RV GLS of 21.1%).

On the other hand, venous congestion evaluation (IVC% and PVP) did not differ significantly between groups, nor was it correlated with acute kidney injury, an association we previously demonstrated in a small series [56]. This could be due to the low incidence of AKI in this cohort.

We should highlight that remdesivir was given in only one-third of our patients, and no difference was found between groups. On the other hand, all patients had been under corticoterapy at some time point. Therefore, these are factors with weak discriminatory power to conclude from any association with their administration.

Within our limitations, it is also worth mentioning the small sample size that could have underpowered our study for more clinically significant outcomes such as mortality, the need for mechanical ventilation, shock, ICU and hospital length of stay.

Moreover, from a clinical point of view, the high rate of cardiovascular risk factors could also point out that pre-existing cardiac conditions of the patients studied may have determined subclinical systo-diastolic dysfunction independent from their ongoing SARS-CoV-2 infection at the time of the study. All the patients in the DYS group, except three who had no registries, had their clinical files checked to confirm the absence of cardiac dysfunction in previous echocardiograms,. Nevertheless, the repeated measures along the time course may have partially overcome this issue for the other results. Despite the extensive STE study, uncommon in other studies, we did not report the EKG abnormalities, which could enrich our population’s spectrum of cardiac disease. Moreover, GLS measurements have been shown to be software-dependent [57]. Therefore, we cannot exclude that our findings may have been different if we had used another commercially available speckle-tracking software. Importantly, experienced operators performed echocardiographic evaluations, which may explain why the intraobserver reproducibility of measurements was better for classic indices than for GLS measurements [58].

Regarding the immunological analysis, we should highlight different sample numbers between the three time points, mostly due to patients’ discharge or death, and some practical limitations in assuring immunophenotypic assays in all samples. However, this was a longitudinal assessment in which each patient was evaluated continuously in subsequent time points, thus ensuring its control over time. Furthermore, only two trained operators processed and analyzed all samples, thus reducing a potential interference in reproducibility generally observed in a multioperator approach. Still, despite the extensive flow cytometric analysis being performed, functional assessments would have brought extra input on the characterization of the analyzed cells and their role in the disease. Finally, IL-1β levels were inexpressive, eventually due to the low sensitivity of the kit used. Nevertheless, we aim to further explore other biomarkers, such as the IL-6 receptor, which may bring additional insights into the immune background of COVID-19 patients, particularly those with associated cardiac injury.

## 5. Conclusions

Our study suggests a characteristic immunological profile in severe COVID-19 patients with cardiac dysfunction (echocardiographically proven) and highlights the importance of early and repeated assessment of LV diastolic function.

Moreover, in this group of patients, a more expressive effector memory T cell compartment at baseline, enriched in activated CD8 T cells, may suggest a persistent inflammatory state with its possible migration to peripheral tissues (such as the heart). Further studies are required to define immune-mediated tissue injury and identify specific host-directed therapy targets for the treatment of this disease.

This study is a pioneer on the extensive longitudinal cardio-immunologic assessment of severe COVID-19 patients, and the associations found can further impact the COVID-19 pneumonia monitorization.

## Figures and Tables

**Figure 1 jcm-11-01880-f001:**
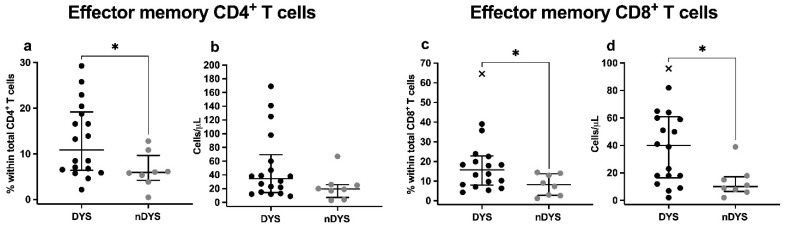
Median percentages or absolute counts with IQR of: (**a**,**b**) CD4^+^ Effector Memory T cells; and (**c**,**d**) CD8^+^ Effector Memory T cells, in patients with (DYS) and without (nDYS) cardiac dysfunction at admission. Differences between DYS and nDYS groups were analyzed with the Mann–Whitney test. * *p*-value < 0.05.

**Figure 2 jcm-11-01880-f002:**
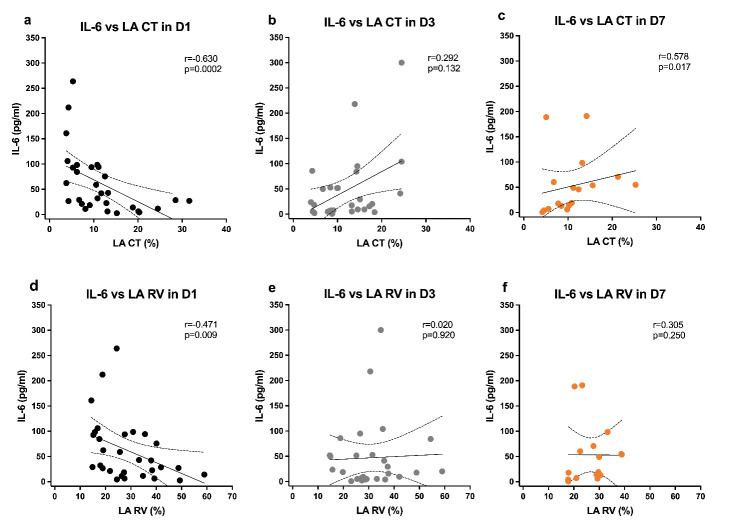
IL-6 levels correlations with LA CT and LA RV in all patients at time point D1 (**a**,**d**); D3 (**b**,**e**); and D7 (**c**,**f**); Spearman correlation coefficients, 95% confidence interval, and *p*-values are indicated. LA CT, left atrium strain contraction function; LA RV, left atrium strain reservoir function.

**Figure 3 jcm-11-01880-f003:**
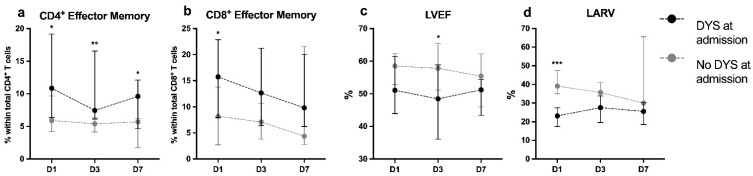
Median percentages with IQR of: (**a**) CD4^+^ effector memory T cells; (**b**) CD8^+^ effector memory T cells; (**c**) left ventricle Eeection fraction; and (**d**) left atrium reservoir strain at admission (D1); at 2nd to 3rd day (D3); and at 7th to 10th day (D7); Black and grey circles indicate groups of patients with (DYS) or without (nDYS) cardiac dysfunction at admission, respectively. Differences between DYS and nDYS groups at each timepoint were analyzed with the Mann–Whitney test. * *p*-value < 0.05; ** *p*-value < 0.01; or *** *p*-value < 0.001.

**Table 1 jcm-11-01880-t001:** Baseline characteristics of patients with (DYS) and without (nDYS) cardiac dysfunction at ICU admission.

	All Patients	DYS	nDYS	*p*-Value
	(n = 30)	(n = 22)	(n = 8)	
Demographics				
Sex, F/M	11/19	7/15	4/4	0.417 *
Age, mean (SD)	60.7 (14.8)	61.0 (15.6)	60 (13.3)	0.870 ^#^
Hypertension, n (%)	22 (73.3)	17 (77.3)	5 (62.5)	0.643 *
DM, n (%)	9 (30.0)	7 (31.8)	2 (25.0)	1.000 *
COPD n (%)	3 (10.0)	3 (13.6)	0 (0.0)	0.545 *
Remdesivir, n (%)	10 (33.3)	8 (36.4)	2 (25.0)	0.682 *
ACEi or ARB, n (%)	15 (50.0)	12 (54.5)	3 (37.5)	0.682 *
BMI, mean (SD)	31.2 (6.1)	31.6 (6.1)	29.9 (6.1)	0.524 ^#^
Cardiac dysfunction				
LV Systolic dysfunction, n (%)	12 (40.0)	12 (55.0)	-	-
Classic criteria	-	10 (33.0)	-	1.000 *
Strain criteria	-	9 (30.0)	-	
LV Diastolic dysfunction, n (%)	19 (63.3)	19 (86.0)	-	-
Classic criteria -grade 1 -grade 2 -grade 3	-	7 (23.3) 4 (13.3) 3 (10) 0 (0)	-	**0.008 ***
Strain criteria	-	18 (60.0)	-	
RV dysfunction, n (%)	11 (36.7)	11 (50.0)	-	-
Classic criteria	-	11 (36.7)	-	0.785 *
Strain criteria	-	9 (30.0)	-	
Morbidity and mortality				
Mechanical Ventilation, n (%)	7 (23.3)	4 (18.2)	3 (37.5)	0.345 *
P/F, median (IQR)	156 (133–207)	157 (131–215)	155 (130–196)	0.793 *
Shock, n (%)	7 (23.3)	4 (18.2)	3 (37.5)	0.345 *
AKI, n (%)	7 (23.3)	6 (27.3)	1 (12.5)	0.638 *
ICU length of stay, median of days (IQR)	8 (5–12)	8 (5–11)	8 (4–13)	0.098 ^$^
Hospital length of stay, median of days (IQR)	21 (13–30)	21 (12–29)	19 (13–37)	0.972 ^$^
Mortality, n (%)	5 (16.7)	3 (13.6)	2 (25.0)	0.589 *
Biomarkers, median (IQR)				
Troponin (ng/L)	13.0 (13.0–18.9)	13.0 (13.0–25.0)	15.5 (13.0–17.0)	0.371 ^$^
Troponin levels above 99th percentile, n (%)	1 (3.0)	1 (4.5)	1 (12.5)	-
NTproBNP (pg/mL)	275.0 (108.0–946.8)	275 (91.5–913.8)	401 (201–1058)	0.468 ^$^
IL-1β (pg/mL)	1.0 (1.0–1.0)	1.0 (1.0–1.0)	1.0 (1.0–1.0)	1.000 ^$^
IL-6 (pg/mL)	37.2 (17.5–93.9)	37.5 (20.6–94.9)	35.4 (12.2–89.7)	0.629 ^$^
ADM (ng/mL)	6.2 (4.3–7.2)	6.4 (5.0–8.5)	5.5 (4.2–6.7)	0.534 ^$^
CRP (mg/dL)	9.6 (5.9–20.3)	9.6 (5.9–17.6)	11.6 (4.8–28.0)	0.872 ^$^
PCT (ng/mL)	0.2 (0.1–0.6)	0.2 (0.1–0.6)	0.1 (0.1–0.4)	0.314 ^$^
ESR (mm/1st h)	120 (88–120)	116 (88–120)	120 (76–120)	0.831 ^$^
Ferritin (ng/mL)	867 (624–1336)	782 (623–1182)	1209 (674–1583)	0.277 ^$^
D-dimers (µg/mL)	0.4 (0.3–1.0)	0.4 (0.3–1.0)	0.5 (0.3–0.9)	0.541 ^$^
Fibrinogen (mg/dL)	634 (526–880)	625 (519–849)	644 (538–976)	0.636 ^$^
Platelets (×10^9^/L)	282 (197–338)	253 (166–326)	321 (266–404)	0.058 ^$^

* Fisher’s exact test: ^#^ Unpaired *t*-test with Welch’s correction; ^$^ Mann–Whitney test. All significant results are indicated in bold: SD, standard deviation; IQR, interquartile range; DM, diabetes mellitus; COPD, chronic obstructive pulmonary disease; ACEi, angiotensin-converting enzyme inhibitor; ARB, angiotensin receptor block; BMI, body mass index; LV, left ventricle; RV, right ventricle; P/F, PaO2 (oxygen partial pressure) to FiO2 (inspired fraction of oxygen) ratio; AKI, acute kidney injury; ICU, intensive care unit; NTproBNP, NT terminal of the prohormone brain natriuretic peptide; IL-1β, interleukin 1 beta; IL-6, interleukin 6; ADM, adrenomedullin; CRP, C-reactive protein; PCT, procalcitonin; ESR, erythrocyte sedimentation ratio.

**Table 2 jcm-11-01880-t002:** Echocardiographic evaluation of patients with (DYS) and without (nDYS) cardiac dysfunction at ICU admission.

Echocardiographic Parameters, Median (IQR)	All Patients	DYS	nDYS	*p*-Value
	(n = 30)	(n = 22)	(n = 8)	
LV EF (%)	53.1 (46.3–62.2)	51.1 (44.0–61.5)	58.5 (53.9–2.4)	0.105 ^$^
LV GLS (%)	25.4 (20.5–29.2)	25.4 (19.5–29.4)	25.4 (22.0–27.8)	0.872 ^$^
MAPSE (mm)	16.4 (15.5–18.6)	17.1 (15.6–19.0)	16.0 (14.5–17.5)	0.197 ^$^
LV S’ (cm/s)	11.0 (9.7–14.2)	11.5 (9.7–14.6)	10.6 (8.9–13.3)	0.314 ^$^
LA CT (%)	10.7 (6.0–13.7)	7.7 (5.1–12.9)	13.9 (11.2–23.1)	**0.0002** ^$^
LA RV (%)	26.9 (18.6–38.1)	23.1 (17.4–27.5)	39.1 (35.0–47.5)	**0.008** ^$^
LV E (cm/s)	71.3 (59.8–82.7)	70.0 (60.8–88.7)	71.3 (56.9–80.3)	0.704 ^$^
LV A (cm/s)	63.2 (55.4–84.4)	60.6 (53.3–88.4)	63.5 (60.0–67.7)	0.973 ^$^
LV E/A	1.1 (0.8–1.4)	1.0 (0.8–1.5)	1.1 (0.9–1.3)	0.933 ^$^
LV E/e’	7.1 (6.6–8.5)	7.6 (6.3–9.3)	6.8 (6.7–8.2)	0.542 ^$^
LA volume (mL)	63.3 (49.6–81.0)	63.3 (49.6–79.6)	62.5 (30.6–95.6)	0.982 ^$^
RV FAC (%)	44.3 (34.0–52.6)	41.5 (31.5–48.1)	53.2 (48.0–59.3)	**0.005** ^$^
TAPSE (mm)	22.0 (19.9–25.1)	21.4 (19.4–23.6)	25.1 (21.1–27.4)	0.089 ^$^
RV S’ (cm/s)	14.7 (12.8–17.3)	13.9 (12.6–17.3)	15.7 (13.9–19.6)	0.282 ^$^
RV GLS (%)	22.9 (18.7–29.6)	21.1 (17.0–27.4)	26.9 (22.7–36.4)	0.062 ^$^
sPAP (mmHg)	30.4 (23.2–39.6)	30.4 (21.8–43.3)	32.6 (23.7–37.2)	0.972 ^$^
IVC (%)	23.5 (10.6–36.9)	23.6 (11.1–38.7)	21.8 (1.9–36.8)	0.467 ^$^
PVP (%)	35.5 (29.4–43.3)	32.4 (27.5–41.0)	41.4 (32.9–48.2)	0.089 ^$^
CI (L/min/m^2^), mean (SD)	2.8 (0.7)	2.8 (0.7)	2.9 (0.6)	0.743 ^#^

^#^ Unpaired *t*-test with Welch’s correction; ^$^ Mann–Whitney test. All significant results are indicated in bold: LV, left ventricle; LVEF, LV ejection fraction; LV GLS, LV global longitudinal strain; MAPSE, mitral annular plane systolic excursion; LV S’, LV tissue Doppler S wave; LA, left atrium; LA CT, LA strain contraction function; LA RV, LA strain reservoir function; LV E, LV early diastolic filling wave; LV A, LV atrial contraction wave; LV E/A, LV E wave to A wave ratio; LV E/e’, LV E wave to tissue Doppler ‘e’ wave ratio; RV, right ventricle; RV FAC, RV fractional area change; TAPSE, tricuspid annular plane systolic excursion; RV S’, RV tissue Doppler S wave; RV GLS, RV global longitudinal strain; sPAP, systolic pulmonary artery pressure; IVC %, inferior vena cava variability; PVP, portal vein pulsatility index; CI, cardiac index.

## Data Availability

The datasets used and/or analyzed during the current study are available from the corresponding author on request.

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
