# Peer review of "Characteristic Immune Dynamics in COVID-19 Patients with Cardiac Dysfunction"

_jcm, 2022, doi:10.3390/jcm11071880_

Round 1

Reviewer 1 Report

It is a well written and interesting article which suggests an association between the immune profile and cardiac dysfunction in severe COVID-19 patients. 

In the pandemic era research that focuses on the underlying mechanisms of action of the new coronovirus is of high clinical impact. 

The authors extensively describe their methods and results and adequately refer to their limitations. Indeed, the small sample size underpowers their study which however gives insight into the immunological profile of COVID-19 patients with cardiac dysfunction.

Author Response

We sincerely acknowledge your reviews and comments. Thank you for your insight and contribution to this manuscript.

Warm regards,

Filipe Gonzalez

Reviewer 2 Report

The authors well describe an interesting association between the immunological trait and cardiac dysfunction in severe COVID-19 patients.

They evaluate  the immune parameters in COVID-19 patients admitted to the intensive care unit (ICU) and  identify distinctive features in patients with cardiac injury.

30 COVID-19 patients >18 years admitted to the ICU were studied on days D1, D3 and D7 after admission. Cardiac function was assessed using speckle tracking echocardiography (STE). Peripheral blood immunophenotyping, cardiac (pro-BNP; troponin) and inflammatory biomarkers were simultaneously evaluated.

The authors concludes that cardiac dysfunction (DYS) was detected by STE in 73% of patients: 40% left ventricle (LV) systolic dysfunction, 60% LV diastolic dysfunction, 37% right ventricle systolic dysfunction. High- sensitivity cardiac troponin (hs-cTn) was detectable in 43.3% of the patients with a median value of 13.00 ng/L.  No significant differences between DYS and nDYS patients regarding mortality, organ dysfunction, and cardiac (including hs-cTn) or inflammatory biomarkers. Patients with DYS showed persistently lower lymphocyte counts (median 896 [661-1837] cells/µl vs 2141 [924-3306] cells/µl, p=0.058), activated CD3 (median 85 [66-170] cells/µl vs 186 [142-259] cells/µl, p=0.047) and CD4 T cells (median 33 [28-40] cells/µl vs 63 [48-79] cells/µl, p=0.005), and higher effector memory T cells (TEM) at baseline (CD4%: 10.9 [6.4-19.2] vs 5.9 [4.2-12.8], p=0.025; CD8%: 15.7 [7.9-22.8] vs 8.1 [7.7-13.7], p=0.035; CD8 counts: 40 cells/µl [17-61] vs 10 cells/µl [7-17], p=0.011) than patients without cardiac dysfunction. 

Congratulation with these conclusion very important for scientific community.

Author Response

(The authors gave the same response as above.)

Reviewer 3 Report

I read this paper with interest because a multi marker evaluation of the myocardial injury is essential in Covid-19 patients. Although the study group is small, the conclusion sounds very interesting. However, it has some important limitations. See below:

  1. It is important to justify clearly ethical aspects, since some patients from the study group are intubated, with severe mental alteration, low oxygen saturation, etc.
  2. The introduction is too long, it looks like a discussion section. It should be revised.
  3. The number of patients is small, in SR and in AFiB? Did they use STE in AFib? It is well known that the STE software might not generate BE for Global deformation parameter if detects a HR variability > 10/BPM. How did they do STE in this cases - please explain.
  4. The authors used speckle tracking echocardiography, a very promising echo technique, in a group of patients in ICU. It should be noted that this technique is extremely dependent of breath hold, hemodynamic conditions (please see the guidelines for STE evaluation), position, etc. All patients from the study had significant respiratory alteration generated by Covid-19. This means that they are not able to hold their breath, for STE evaluation. Please explain. 
  5. In this light, the author have to present one Figure with STE evaluation for LV, LA, RV function in a invasive ventilated patient, and also a clear table for variability inter and intra observer for all STE parameters, not only a general sentence, as is in the current manuscript. STE might be altered by breath frequency, and the conclusion about low deformation for LV , LA and RV parameters might be altered.
  6. The examination position in STE is also essential. Please explain clearly the position in IOT patients, in CPAP patients, etc (see also the guideline for STE evaluation).

Author Response

We sincerely acknowledge your reviews and comments. Thank you for your insight and contribution to this manuscript.

Replying to your comments:

  1. According to local legislation, consent from the patient or next of kin to record data and draw blood was obtained before study enrolment. In the event that informed consent cannot be provided due to incapacity at the time of data collection, it will be placed at a legal representative; the research team will provide informed consent to patients who acquire this ability so that they can validate the use of your data. If the patient dies before there has been
    consent, this will be requested from a legal representative. For the data
    already collected in these cases, as they are non-sensible (anonymized) data from the patient, the ethics committee would be asked to include them in the study.

2. Thank you for your suggestion. We already shortened the introduction.

3. We had only 2 patients in AFib and they were excluded not by the AFib, but by the limited acoustic window. For the study design and preparation, we checked the STE software from Aplio Xario which allowed AFib and we expected to do 10 measurements for each parameter in these patients. 

4. In ventilated patients we were able to do manual inspiratory holds for a better acoustic window and STE detection (inclusively this is a manoeuvre used to assess and calculate respiratory mechanics).

5. Thank you for your consideration. You're absolutely right for the variation problems, which gave us a lot of hard work to get the STE well done. For the inter-observer variability (median (IQR 25-75)): LV GLS 0.8 (0.5-1.1), RV GLS 0.7 (0.5-1.1), LA RV 1.3 (0.9-1.9), LA CT 1.1 (0.8-1.7). For the intra-observer variability: LV GLS 0.6 (0.5-0.8), RV GLS 0.7 (0.5-0.8), LA RV 1.3 (0.5-0.9), LA CT 0.9 (0.4-0.9).

I'll ask the editor to upload a figure of each.

6. With the help of nurses, we positioned every patient in the left lateral decubitus previously, for the best acoustic windows possible.

Once again, thank you for your comments and suggestions. We look forward to knowing your feedback. 

Warm regards,

Filipe Gonzalez

Reviewer 4 Report

The manuscript by Filipe et al. studies the immune profiling in the COVID-19 patients admitted to intensive care. The authors collected the cardiac data with the measurement of echo and BNP and very interestingly, they evaluated inflammatory markers in peripheral blood. This manuscript suggests a correlation between the immunological traits with the cardiac function during COVID-19 infection.  Overall, the work is novel and can be considered for the readership of JCM after improving it.

  1. This manuscript is about an association with the dynamics of immune parameters and cardiac function during the COVID-19 infection regimen. It would be more appropriate if the authors could include some introductory parts in the introduction section that discuss how inflammation dynamics affect cardiac function.
  2. The authors examined the percentage of CD4 and CD8 T cells in DYS and nDYS groups and as expected they are significantly lower in the nDYS group. The authors need to clarify that proliferation of CD4+ and CD8+ T cells are downregulating with the clearance of Antigens or the drug treatment during the infection regimen inducing apoptosis in these cells.  
  3. The inflammatory cytokine IL-6 mostly comes from the side of the myeloid cell, as authors show correlation diagram of IL-6 therefore authors need to discuss the status of myeloid cells in these groups.
  4. It would be more appropriate if authors could include the gating strategy of flow data, how they calculated the percentage of Cells.

Author Response

We sincerely acknowledge your reviews and comments. Thank you for your insight and contribution to this manuscript.

Replying to your comments:

1- Regarding the appropriate background on inflammation and cardiac dysfunction, we agree with you, and we already had a more robust introduction and referencing for this matter. But all the other reviewers repeatedly comment on having a lengthy introduction and cutting it as much as possible. We've been enriching the discussion with the topics that reviewers regard as important to compensate for this. If you still consider it insufficient, we can reach an agreement with the Editor to add this specific content to the introduction.

2- We understand the relevance of this point, and we thank the reviewer for this comment.  

We have clarified this point in the discussion section.

[Discussion, page 13, line 392]

Furthermore, subsets may be proliferating even though they are decreased in frequency 24, with proliferation rates likely to be influenced by disease stage and severity and even enhanced in patients with severe disease 26. Additionally, the increased presence of effector cells in the DYS group may also reflect a later and less effective antigen clearance, with increased stimulatory signals and proliferation rates leading to exhaustion27. On the contrary, patients without dysfunction may experience a reduction in antigen levels, consequently leading to increased apoptosis of antigen-specific effector cells in an attempt to achieve immune homeostasis. Thus, the impairment in the clearance of inflammatory signals and differentiated cells could be related to the status of cardiac dysfunction. Moreover, reduced antigen stimulation can also be an effect of the ongoing treatment with anti-inflammatory agents REF- DOI: 10.1007/s00018-005-5390-y. In brief, the underlying mechanisms for increased levels of effector/memory T cells in DYS patients remain unclear; nonetheless, they most probably result from the conjugation of altered differentiation patterns, proliferation-apoptosis imbalance, and even deviated migratory events between periphery and the affected organs and tissues 27.

3- We acknowledge the reviewer's comment, and we recognize that discussing the status of myeloid cells would improve our manuscript. Thus, the paragraph on IL-6 and the interplay with myeloid cells were thoroughly revised and supplemented.

[Discussion, page 13, line 405]

In discussion: Interestingly, although our patients had increased serum levels of IL-6 7,8,16,28, in accordance with previous studies, the cytokine levels were not different between DYS and nDYS patients. Accordingly, circulating neutrophils and monocytes, major sources of IL-6 during inflammatory responses, showed no important variations between both groups. NDYS patients had an increase of circulating monocyte counts seven days after admission, with a significant decrease of serum PCR at the same time point, possibly supporting an earlier recovery of inflammation in nDYS (though IL-6 levels didn't change accordingly). Moreover, IL-6 levels were not correlated with neutrophils and monocytes counts at any time point. Of course, we cannot obliviate other sources for this cytokine, such as endothelial cells, particularly when endothelial activation and disruption have been considered major drivers of severe COVID-19 Ref. https://doi.org/10.1038/s41392-021-00819-6. Nonetheless, the absence of differences on IL-6 between DYS and nDYS can result from the impact of ongoing therapy since approximately half of the patients were under ACEi in both groups, and these agents are known to have an impact on both immune cells and serum cytokines, like IL-6 Ref. DOI: 10.3389/fcvm.2021.710946.  

Still, left atrial strain function parameters, such as LA RV and LA CT - a more sensitive surrogate for diastolic dysfunction - correlated negatively with IL-6 levels at admission, which could help identify patients at risk for a protracted inflammatory state. A previous systematic review defied the role of IL-6 in COVID-19 29.

4- We accept the reviewer's suggestion and acknowledge that the gating strategy included in "Additional file 2" could be improved. Despite the legend of the figure discriminates how the sequential analysis was performed and the mother populations considered for each analyzed subset (i.e., against which populations the percentage were calculated for each subset), we have clarified this in the figure and the respective legend.

Please find the following changes in the revised manuscript:

[Additional file 2 legend, page 18, 613]

"(…) (g, I, k) T cell major populations were divided into CD4, CD8, and gd T cells. (h, j, l) cCharacterization of naïve (N), central memory (CM), effector memory (EM) and terminally differentiated effector (TD) cell subsets within total CD4, CD8, and gd T cell populations. (m) All activated the aforementioned T cell subsets were evaluated with for HLA-DR expression (activated cells). (n) The following subsets were identified within total B cells based on IgD/CD27 markers as naïve, unswitched and switched memory, and double-negative cells (DN). (o, p) Transitional B cells and plasmablasts were further characterized using CD38, CD27, and CD20 markers. (q) Identification of IgM+ positive plasmablasts within total plasmablasts. Absolute counts for the major leucocyte and lymphocyte subsets were obtained directly from counting beads. Percentages for each subset were calculated within the respective mother population."

We're available for any remaining questions.

Warm regards,

Filipe Gonzalez

Round 2

Reviewer 3 Report

Thank you for all clarifications.